# Decarbonization Tradeoffs: A Dynamic General Equilibrium Modeling Analysis for the Chilean Power Sector

**Shahriyar Nasirov \*** , **Raúl O'Ryan and Héctor Osorio**

Facultad de Ingeniería y Ciencias, Universidad Adolfo Ibáñez, Avenida Diagonal Las Torres 2640,
7941169 Peñalolén, Santiago, Chile; raul.oryan@uai.cl (R.O.); heosorio@alumnos.uai.cl (H.O.)
\* Correspondence: Shahriyar.nasirov@uai.cl

**Abstract:** Medium size developing countries like Chile that commit to decarbonization goals need to carefully assess the trade-offs associated to their intensity and timing, since most of the technologies required will be absorbed, not produced, by these countries. A rapid expansion of renewables in the Chilean energy matrix, mostly thanks to exceptional solar and wind resources, combined with a rapid decrease in the cost of renewable energy technologies, intensified current policy debates to reduce the role of coal, which is the largest source of $CO_2$ emissions in the generation mix. Recently, the main generation companies in Chile made a voluntary commitment to not invest in new coal projects that do not include carbon capture and storage systems. In addition, the Chilean government announced its plans to phase out coal plants completely by 2040. In this context, the aim of this research is to study the economy-wide and emission reduction impacts of different decarbonization paths in the Chilean power sector. For this purpose, we consider dynamic simulations using a new energy-oriented version of the Computable General Equilibrium Model (CGE)- General Equilibrium Model for the Chilean Economy (ECOGEM)-Chile which is soft linked to the bottom-up engineering energy model. The results show the major impacts under both the business as usual (BAU) scenario and the coal phase-out scenario. Additionally, the study discusses to what extent the ambitious decarbonization goals of the Chilean government are coherent with the current technological limitations.

**Keywords:** dynamic CGE models; decarbonization; Chile; power sector

---

## 1. Introduction

The Paris Agreement pushes countries to propose increasing ambitions related to $CO_2$ emission mitigation. Developing countries must carefully assess these commitments ex-ante, especially those relating to the energy sector, since the resources required may compete with other development goals defined in the 2030 Agenda for Sustainable Development with importance of the energy sector in emissions of developing countries) [1]. A complete study of the links between the 169 declared sustainable development goals (SDGs) and actions relating to SDG7—ensure access to affordable, reliable, sustainable, and modern energy for all—identifies 65 trade-offs, nearly all related "to the tension between the need for rapid action to address key issues for human well-being (for example, poverty eradication, access to clean water, food, and modern energy, and so on), and the careful planning needed to achieve efficient energy systems with a high integration of renewable energy" [2]. Many other studies [3–5] conclude that there is a need to better connect and extend the evidence on trade-offs and synergies of the relation between renewable energy development and advancing towards sustainable development.

There are multiple modelling approaches that allow for the systematic assessment of the trade-offs between renewable energy development and other societal goals. Horschig et al. [6] reviews the

strengths and weaknesses of three quantitative (ex-ante), three qualitative (ex-post), and more recent hybrid approaches. Ex-ante evaluations include input-Output (I/O) modeling, computable general equilibrium modeling, and system dynamics modeling, models that are more useful for policy evaluations because they are inter-sectoral and distinguish different options of production and consumption [7]. They conclude that CGE modeling has its strengths in economy-related policy evaluation issues—that relate to different SDGs—and for analyzing long term effects, but it struggles with a usually low level of technical detail. Besides, over the last decade there are growing number of literatures using CGE models to examine the impact of renewable energy development [8–11].

To overcome in part the lack of technological detail, hybrid modelling approaches that combine different models are now currently being used, in particular technology rich "bottom-up energy systems models" or engineering models together with "top-down" CGE models. The literature distinguishes in this case between hard and soft linking of models [12–14]. Usually, under soft linking, the information transfer between the models is directly controlled by the user, whereas in a hard-link approach integration is carried out without any user control. In a recent study, Delzeti et al. [15] distinguish between one-way linkage, in which outputs from one model (i.e., the engineering model) are used as exogenous parameters or variables in the other model (i.e., the CGE model), and a two-way linkage, that considers the feedback between models to obtain better convergence of similar variables. After a review of the literature, they propose that one-way linking is sufficient if the focus is on an economy wide picture based on a given pathway/constraints. In this context, one-way linkage seems to be an adequate approach to examine different decarbonization paths for the energy sector and the impact on some relevant SDG variables.

Over the past few years, the Chilean power sector has experienced rapid changes with the extraordinary growth of Non-Conventional Renewable Energy Sources (NCREs) in the energy matrix. The share of NCREs in the total installed energy capacity increased from only 5% in 2014 to 23% by May 2020. An accumulative investment in large-scale renewable energy projects amounted to $14.8 billion between 2010–2019 [16], making the country one of the most attractive clean energy markets in the region. This has been possible thanks to the country's exceptional resources, the rapid reduction of costs of renewable energy technologies and coherent policy frameworks. Chile ranked second in the Climatescope survey in 2018 [17], thanks to its overall attractiveness for foreign investment, solid public policies on clean energy and a proven track record of overall economic stability. As of 2019, NCRE projects with environmental approval accounted for 33 GW, which is even higher than the total current installed capacity of the power sector [18]. This shows a promising prospective for these technologies in the country's future energy matrix.

Increasingly ambitious targets for renewable energy combined with rapidly falling technology prices could result in record installations of these technologies in the near future. The promising future of renewables encouraged the government to expand its efforts to decarbonize the power sector by phasing out its fossil-fuel infrastructure. In terms of overall greenhouse gas (GHG) emissions, 78% of $CO_2$ emissions in Chile comes from the energy sector, where the power sector is the main contributor with 41.5%. For this reason, the power sector remains the main policy focus for the implementation and compliance of the NDC in the rapid decarbonization. In particular, coal-fired power plants have been considered to have one of the largest potentials, as this sector emits the largest share of emissions. In mid-2019, the government announced its plans to close 8 (out of 28) older coal-fired power plants by 2024, and the rest by 2040. Moreover, the government has adopted recent ambitious targets in its National Energy Policy 2050, aiming to meet 60% of national electricity generation with renewable energy by 2035 and 70% by 2050.

Although examining the impact of renewable energy development using a dynamic CGE model has been tested in the context of various countries, mostly developed ones, a few studies have been found about Chile and the South American region. For Chile, a previous study by the authors [8] presents an initial one-sided soft link model using the general equilibrium General Equilibrium Model for the Chilean Economy (ECOGEM)-Chile model to build a more realistic baseline scenario for Chile

up to 2050. This is a recursive dynamic model with a significant sectoral, labor market, and external market disaggregation. Sixty sectors are included in the dynamic version, and to better characterize the expected development of the electricity generation sector, it considers seven different subsectors (coal, hydroelectric, solar, wind, gas, oil, and wood). Using a soft link with an engineering model of the energy sector, a baseline has been proposed up to 2050 for Chile. This is a novel approach for Latin American countries and mid-sized developing countries in general. Another recent paper for Chile by [19] uses a general equilibrium approach to examine green growth opportunities for Chile. Using a 15-sector dynamic stochastic general equilibrium (DSGE) model that includes non-renewable and renewable electricity generation, they examine the macroeconomic changes due to a mitigation package with 15 measures proposed for Chile, including coal phase out. This DSGE model cannot simulate the detailed sectoral impact of the ECOGEM model because it has only one electricity generation sector so it does not incorporate solar and wind generation separately. It does not incorporate the expected dynamics of these two key electricity generation sectors, nor the potential impact of this on the baseline up to 2050. Its main aim is to model the impact of the $CO_2$ mitigation intervention package on the economy, particularly for aggregate macroeconomic indicators.

However, to the best of our knowledge, few similar studies can be found in international literature [20–23] and none at all for the Chilean and Latin American region, focusing on the economy-wide assessment of the decarbonization goals. Besides, an application of the methodology which links the CGE model with an engineering model allows for more realistic technology-based scenarios.

In this context, the aim of this research is to study the economy-wide impacts, particularly the important tradeoffs related to the SDGs under the decarbonization scenario for the Chilean power sector. Additionally, it aims to study the potential impact of decarbonization on the country's $CO_2$ emissions baseline. For this purpose, we developed an advanced version of the computable general equilibrium (CGE) model—ECOGEM-Chile model (General Equilibrium Model for the Chilean Economy). Following best practices, we use the one-way linkage approach to propose a more realistic energy baseline. This baseline is then shocked with different decarbonization options. The main purpose of this study is to develop and implement a climate policy simulation and modelling tool to evaluate policies' economic and environmental impacts to meet the climate change mitigation objectives and to assist decision-makers by offering recommendations for the development of a combination of efficient climate policies. In particular, the development of this tool aims to quantify the economic impact and emissions reductions generated by the different policy alternatives proposed, and/or possible policy combinations, and to help the key private and public stakeholders understand and evaluate the potential impacts of the policy options proposed.

This paper is organized in the following way. Section 2 provides an overview of the Chilean power sector and the role of the coal plants in the generation mix. Section 3 discusses the major features of the newly constructed ECOGEM-Chile model and details how the outputs from a bottom-up engineering model are linked into our ECOGEM-Chile. Section 5 provides the model results for business as usual (BAU) and coal phase-out scenarios and discusses the major variations. Finally, Section 6 offers a discussion on whether or not the ambitious decarbonization goals are coherent with current technological limitations in Chile.

## 2. The Chilean Power Sector: Role of Coal Plants

Chile has been a pioneer in the introduction of comprehensive market reforms to its power sector in the early 1980s [24]. The main objective of these reforms was to break down the traditional structure of the vertically-integrated monopoly into three main segments: generation, transmission, and distribution, and begin the commercialization and privatization of the existing state-owned electricity system. Thus, the role of the State has been minimized, and has only been in charge of performing regulatory and control functions. The Chilean power sector has historically been divided into two major power grids: the Sistema Interconectado del Norte Grande (SING) and the Sistema

Interconectado Central (SIC), reflecting the vertical geographic structure of the country. Electricity demand for the central and southern regions of the country, which represent over 90% of the population, was traditionally supplied by the SIC, while the SING was the main power grid meeting the primary electricity demands for the mining and mineral industries in the north of the country. For decades, the SIC and the SING functioned independently. This structure changed in 2017 when a new project to interconnect the SIC and the SING grids across 600 km was completed. In the place of the SING and the SIC, a new and unique power system called the National Electric System (SEN) was established for the purpose of meeting the energy of more than 97% of the national population. Total electricity installed capacity of the SEN reached 23,315 MW by 2018. Of this capacity, thermoelectricity, conventional large hydro plants, and non-conventional renewable energy technologies (NCREs) represented 53%, 26%, and 21%, respectively. As Chile's economy continues to grow, it is expected that the power system will double its current size, reaching 59GW of installed capacity by 2050 [16]. Regarding gross electricity generation, during the same year, total energy generation amounted to a total of 75,641 GWh in the SEN, which represents 99.3% of the total generation throughout the country. This total is made up of 54.5% thermoelectricity, 28.2% conventional large hydro sources, and 17.4% NCREs.

*Role of Coal Plants in the Generation Mix*

Coal has been considered to be one of the key energy sources in Chile for many decades. Chile has one of the largest coal reserves in South America, which accounts for 4.1 billion tons, mostly in the south of the country [24]. However, due to high exploitation costs, domestic coal demand is met mostly through imports. Although the opening of the Mina Invierno mine in the Magallanes region in 2013 significantly increased domestic production, more than 80% of its total coal supply still depends on external sources. In 2016, net coal imports amounted to 10.6 Mt, more than double the 2006 level. Among the source of imports, Colombia (42% of the total), Australia (29%), and the United States (22%) represent the largest shares.

Over 90% of the total coal supply in Chile is used in electricity generation. Over the last decade, the share of coal consumption in the power sector has been on a general upward trend. It has become the largest and most affordable source to meet growing energy demand, particularly during the energy crisis when Argentina stopped gas exports as of 2004. Over the last decade, the share of the coal consumption in the power sector increased more than twofold from 18% in 2006, reaching 41% in 2016. Figure 1 presents an evaluation of coal's participation in the Chilean power sector between 1976 and 2016.

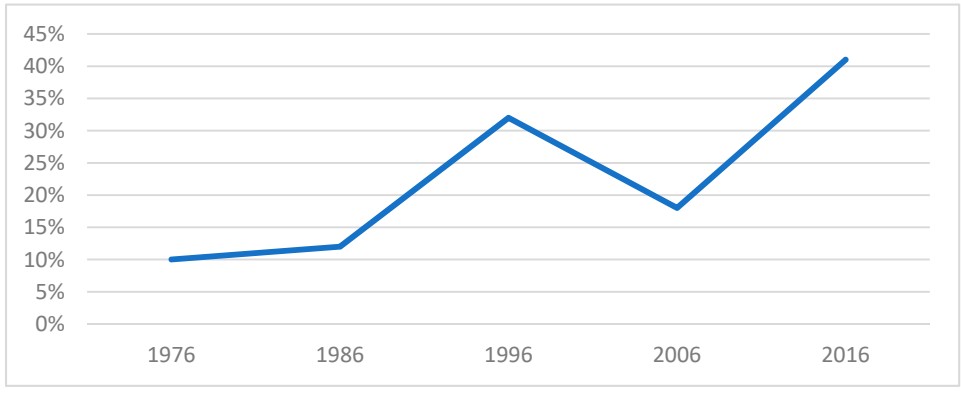

**Figure 1.** Coal in the Chilean power sector, 1976–2016. Source: [24].

Today, Chile obtains more than one-third of its electricity from generation by 28 coal-fired power plants with a generating capacity of 5500 MW. These coal plants are owned by the four largest power companies: the American-owned company AES-Gener with 15 plants, the French-owned Engie with 9 plants, the Italian-owned ENEL with 3 plants, and the Chilean-owned Colbún, owner of one plant.

Coal-fired units provide baseload power to the power system with the highest annual capacity factor of more than 65% for electricity generation in Chile. Although coal may provide some technical and economic benefits for the Chilean electricity system, its increasing use is not consistent with the government's climate policy objectives as it is the main contributor of $CO_2$ emissions in the country. It represents two-thirds of total energy-related emissions in the country. For this reason, environmental opposition to the coal plants in Chile has been strong and effective. The failure of two large coal-fired projects "Barrancones (540 MW)" and "Castilla" (2100 MW) in central Chile are a good example of controversies that provoked highly visible public debates and large community protests [25].

## 3. Methodology—The ECOGEM-Chile Model

In this paper, we developed a new advanced version of the ECOGEM-Chile macroeconomic model in which the power sector is disaggregated. Moreover, given that energy technology analyses and both technical and economic parameters are better modelled and more realistic in the bottom-up-based engineering models, we developed a technique to link the simulation results from these models into our ECOGEM-Chile model. Coupling the bottom-up engineering models and ECOGEM-Chile models allows for a better evaluation of the impacts of technology-based scenarios on the economy-wide and emission assessments. Figure 2 shows how the ECOGEM-Chile model and a bottom-up energy model are coupled. Outputs include $CO_2$ emissions and the economy-wide and sectorial impacts. The methodology is composed of a short overview of the ECOGEM-Chile model, a brief description of the technique linking the bottom-up engineering model, and ECOGEM-Chile model, and an overview of the business as usual (BAU) and coal-fired plant phase-out scenarios.

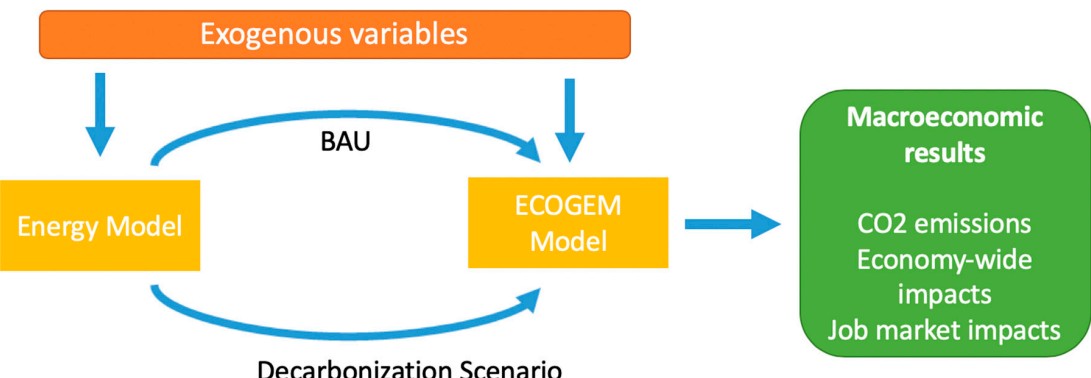

**Figure 2.** Description of General Equilibrium Model for the Chilean Economy (ECOGEM)-Chile.

The ECOGEM-Chile model used in this study is characterized by its multi-sectorial approach, separation of households by income quintiles, breakdown of information by relevant trade partner, and specification of the different production factors, among others. It is a model essentially founded upon neoclassic theory, where savings determine the investment, and it is assumed that there is competitive balance in all markets after a process in which the sectors minimize their costs and maximize their profits. ECOGEM-Chile is a "dynamic recursive" model, meaning that the agents are assumed to be short-sighted and maximize their target functions from period to period, adopted from the original the Organisation for Economic Co-operation and Development (OECD) GREEN model [26–28]. The simulations of each period are linked through the accumulation of capital (investment), which is endogenous, and exogenous suppositions regarding the GDP, job growth, and productivity trends. The model is composed of productive sectors or activities, several occupational categories, income groups (quintiles) for households, public spending categories, final demand spending, trade partners, and different pollution types. The summary of the ECOGEM-Chile model in its current status is presented in Table 1.

**Table 1.** Summary of characteristics of the ECOGEM-Chile model.

| CHARACTERISTIC | DESCRIPTION |
|---|---|
| Sectors | 60 sectors. 27 production sectors (including copper mining); 26 service sectors (excluding electricity generation); 7 electricity generation. |
| Labor categories | 12 categories: high, medium, and low income, by gender (M, F) and urban/rural. |
| Households | Up to 10 deciles |
| Trade partners | 35 trade partners from more important countries (Brazil, USA, China, etc.) and groups of other countries or regions (rest of Asia, Americas, etc.) |
| Public finance | Considers the detailed breakdown of taxes and transfers: direct and indirect corporate taxes, direct household taxes (income tax), employment tax, Duties, a value-added taxes (VAT), government to household transfers, transfers abroad and from abroad. |
| Pollutants | The emissions factors inherent to Chile have been estimated by sectorial production and final consumption |

Source: own elaboration.

The production sectors operate under constant scaled returns. The technology is given by a production function modeled through CES/CET functions (Elasticity of Substitution—Constant Transformation) with a tree structure. Figure 3 below summarizes this tree structure. To obtain the intermediate inputs basket and KEL, fixed proportions are assumed (Leontief-type function). On the factors side, the capital basket is divided into energy and work, using a new CES function, and energy is successively separated from capital, always assuming CES functions for substitution between factors as well as within them (types of work, energy, and capital). The energy basket is divided into electricity and other energies used in production processes that can be substituted through CES.

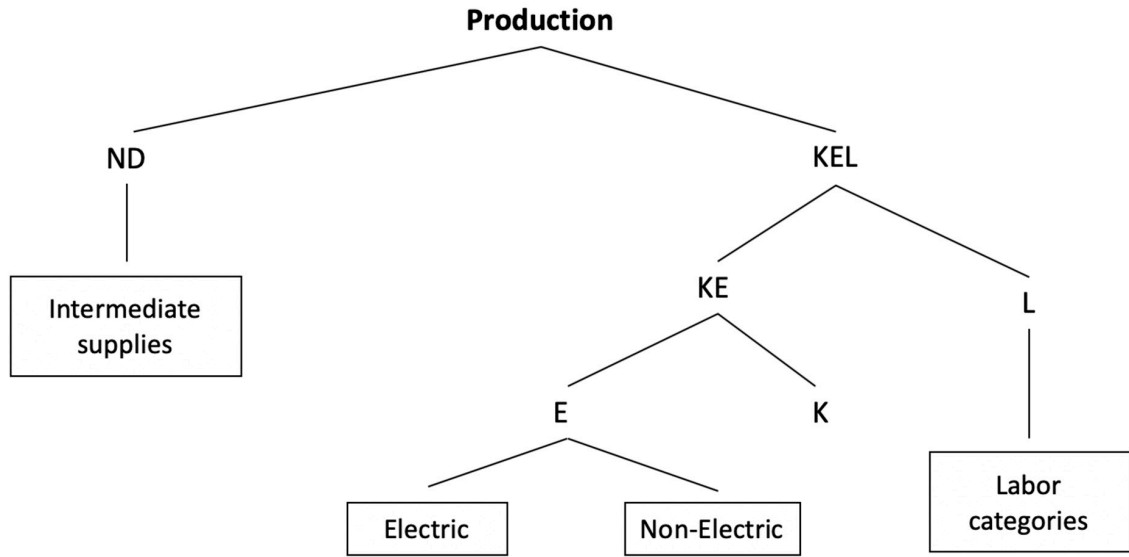

**Figure 3.** Structure of the production tree.

The electric sectors include seven types of electricity generation (gas, coal, fuel, wind, solar, biomass, and hydro power), and other energies include oil, fuels, natural gas, and combustible coal which count as energy inputs.

The production function distinguishes between the use of national and imported products. This difference can be seen in an Armington function, that is, it considers the imperfect substitution between national and imported products. As in the case of production, there is a CES function that

allows for substitution between the national and imported basket. Meanwhile, the national offer receives a similar treatment as that given to demand, now incorporating a CET function to distinguish between the national and exports market.

The model assumes that consumers receive income as owners of different production factors and that part of this income is set aside as savings (proxy for future consumption). The decision between consumption and savings is static, that is, savings are treated as another "good" and determined simultaneously with demands for other goods. Households distribute their income between savings and consumption through an ELES (Extended Linear Expenditure System) profit function. This function also incorporates the minimum consumption of independent subsistence of the income level. Consumers maximize their profits and this maximization is subject to budget restriction.

Once the intermediate demands and those of households are defined, then the rest of the final demands (i.e., investment, government spending, and import and export margins) should be included. The final demand of each item is defined as a fixed portion of the total final demand.

In terms of public finance, this includes different taxes and transfers. The model defines: employment taxes, corporate taxes, income taxes (broken down by quintile), of which are direct taxes. It also defines duties and subsidies on imports, taxes, and subsidies on exports (different by sector) and VAT (national and imported, and by sector). The government also receives and gives direct transfers abroad, has expenses or consumption, and makes transfers to households.

As a closure condition for public finance, the model allows two alternatives: in the first case, government savings are defined as fixed, just like the original level before any simulation, allowing for adjustment through government taxes or transfers. In the second case, government savings are variable, maintaining government spending, tax rates, and transfers as fixed. Apart from the closing rule for the government, where public savings can be determined endogenously or exogenously, as applicable, investment is determined by the savings-investment identity. For this, the value of total demand for private investment is equal to all resources available in the economy for that purpose (retained earnings or company savings, total household savings, capital flows from other countries or external savings, government savings net of expenses in the variation of stock. In this model, external savings is not considered dependent on endogenous variables (e.g., country risk, interest rates, etc.), but rather exogenous ones.

The last closure rule is related to the payments balance equation. Here, the value of imports at international border prices (when assuming a small country, it is not affected by import prices) must be equal to the value of exports at international border prices, plus transfers, the payment of factors, and net capital inputs. Using Walras's Law, the payment balance restriction can be removed from the model (which also allows for its correct calibration).

### 3.1. Coordination of the CGE and Engineering Models

Coordination is done using a method that allows for interaction between the two models through outputs from one model to the other. For this purpose, it is important to determine the common variables or "points of connection" where the models can converge. In this particular study, the electricity generation projections from the energy model were incorporated in the macroeconomic model (CGE). This allows the CGE model to use these projections as a main guide for the evolution of sectors associated with electricity generation. Because this research focuses on the decarbonization of the electric sector primarily under the coal plant phase-out scenario, electricity generation is the main focus of the study on other energy sectors. In particular, two common parameters of the two electricity generation models were chosen to coordinate both models in this study. These include the total quantity of electricity generated and the percentage share of each technology. The intervention

in the CGE model was applied next to the energy demand in production (decomposition of energy bundle) which is represented:

$$XAP_{elec,j} = \sum_{v} aelectp_{j,v,elec} \times elect_{j,v} \times \frac{pelect_{j,v}}{PA_{elec}}^{sigmaelec_{j,v}} \times lambdaep_{j,v}^{sigmaelec_{j,v}-1} \qquad (1)$$

The variable -*aelectp* determines the percentage share of electricity demand in a sector and is automatically adjusted in the model through optimization. The composition of electricity generation in the model contains seven generation technologies. The parameter *lambdak* is the capital efficiency by sector which establishes the capital requirement of a sector in order to produce. At the same level of production, a higher capital efficiency indicates that the capital requirement is lower, which also has a direct effect on the production sector's prices.

The percentage share of each technology in the energy model is directly incorporated into the CGE model through the variable *aeletcp*. For each of these sectors (except for generation) in the CGE model, the electric composition that they demand must be the same composition taken from the energy model, which also implies that the total composition demanded will be equal to the individual composition of each of the sectors, given that the variable determines the proportions demanded and not the total amounts. This manages to replicate the percentage share of each of the generation technologies considered. It must be highlighted that not all electricity generation technologies considered in the energy model could be considered in the CGE model. Some generation technologies from the first model had to be grouped to coincide with the technologies considered in the second model. The technology differences and groupings are summarized in Table 2.

**Table 2.** Considered Technologies.

|  | **Energy Model** | **General Equilibrium Model for the Chilean Economy (ECOGEM)** |
|---|---|---|
| Considered technologies | Solar CSP Solar PV | Solar |
|  | Coal | Coal |
|  | Diesel | Diesel |
|  | Hydro passing Hydro dam Hydro series | Hydro |
|  | Biomass | Biomass |
|  | Geothermal | N/A |
|  | Gas | Gas |
|  | Wind | Wind |

The other parameter, total amount of energy generated from the energy model, was incorporated in the CGE model, modifying the capital efficiency parameter. The parameter establishes the necessary capital requirement of one sector in order to produce. Modifying this parameter (for example, making one sector more expensive or cheaper) helps influence the sectors' prices, which indirectly implies modifying the quantity produced. With this in mind, it was decided that the seven electricity generation sectors found in the CGE model would be modified in the same way. By controlling this parameter, it is possible to achieve total electricity generation values and replicate the electricity generation scenario of the energy model.

### 3.2. Scenario Simulation

Through this coordination, the two energy model scenarios are replicated in the ECOGEM-Chile model. These include the BAU scenario and the coal plant phase-out scenario generated by the energy model and the replications by the ECOGEM-Chile model. The scenarios describe different development paths in the energy generation sector in Chile, and their implications on proportion of renewable sources in the energy portfolio.

The BAU scenario includes the scenario that projects the growth of the energy sector without considering any type of shock, considering the national energy characteristics and trends and the expected evolution of the current energy technologies being used around the world. Considering that the copper sector is the main sector of the Chilean economy in energy consumption, the copper projections were important for the definition of future trends. Given this, it was decided that this production would be given exogenously, and the electricity generation industry would be manipulated exogenously during the coordination process of the two models. Figure 4 shows the electricity generation share trends obtained in the energy model for 2020–2050. On the other hand, Figure 5 represents replications of the results obtained from the energy model in the ECOGEM-Chile model. As shown in Figures 4 and 5, the proportion of non-conventional renewable energy technologies (solar, wind, and biomass) in the energy generation mix under the BAU scenario represents 20% in 2020, 40% in 2030, and 60% in 2050. However, although the weight of coal-fired plant generations has dropped, coal generation continues to play an important role in the Chilean generation mix under the BAU scenario.

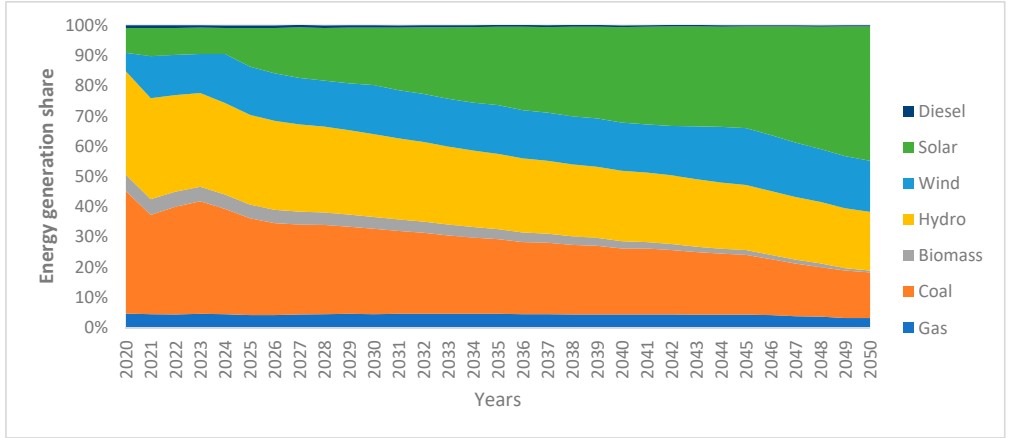

**Figure 4.** Electricity generation share obtained in the engineering model under business as usual (BAU)**.** Source: own elaboration.

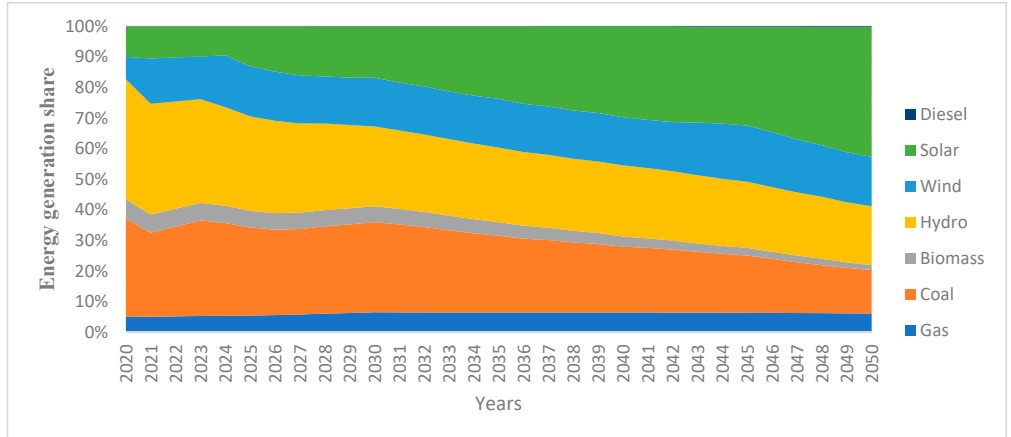

**Figure 5.** Electricity generation share in baseline replicated with Macro model. Source: own elaboration.

Coal Plant Phase-Out Scenario

The coal plant phase-out scenario includes the gradual reduction of coal generation between 2020 and 2050. As shown in Figures 6 and 7, the proportion of coal technologies in the energy generation mix represents 38% in 2020, 20% in 2030, and 0% in 2040. On the other hand, non-conventional renewable energies were gradually replacing carbon over those years.

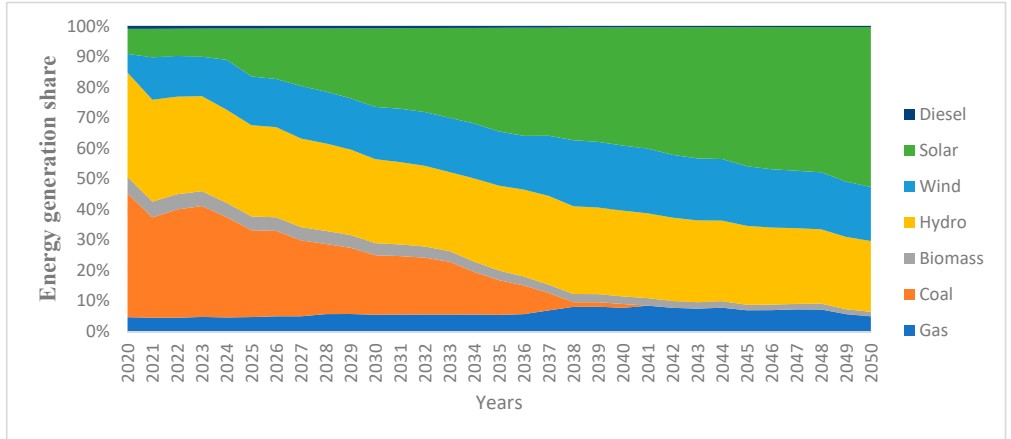

**Figure 6.** Electricity generation share obtained in the engineering model under plant phase-out scenario. Source: own elaboration.

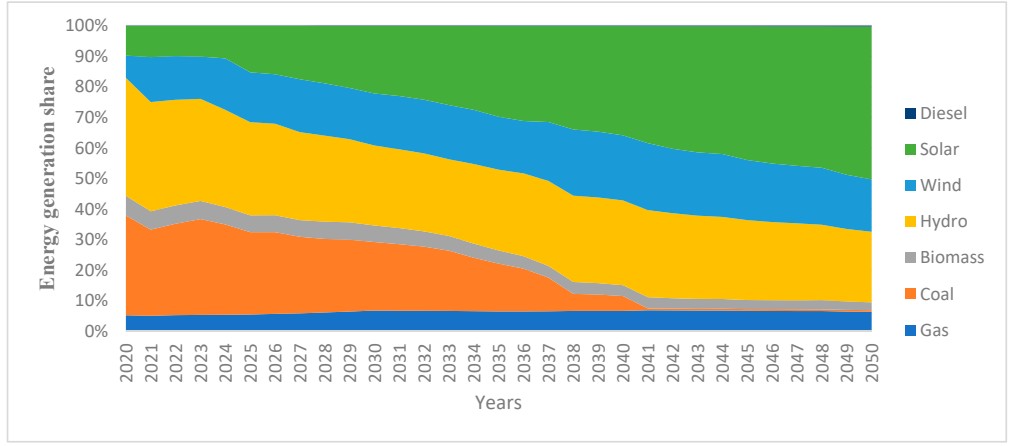

**Figure 7.** Replication of plant phase-out scenario by the ECOGEM-Chile model Source: own elaboration.

## 4. Empirical Results

This section describes the simulation results. First, a BAU baseline and then coal-fired plant phase-out scenario are simulated. The results report the variations (percentage changes) in $CO_2$ emissions, $CO_2$ emissions intensity pathways, and the economy-wide impacts after reducing coal generation in the power sector. Figures 8–10 describe the percentage variations for $CO_2$ emissions and $CO_2$ emissions intensity pathways, respectively, between 2020 and 2050. This is mainly because a transition in the energy matrix from coal-fired plants to renewable energy generation plays an essential role in reducing Chile's $CO_2$ emissions and emissions intensity for 2050. The energy matrix under the BAU scenario still largely relies on coal-fired plants, which are important contributors of $CO_2$ emissions. Currently, the Chilean power system has 28 active coal-fired power plants, providing almost 40% of the country's power, while generating 26% of all greenhouse gases. Decarbonization of the power sector by phasing out coal-fired power plants and replacing them with renewable energies is expected to generate better results.

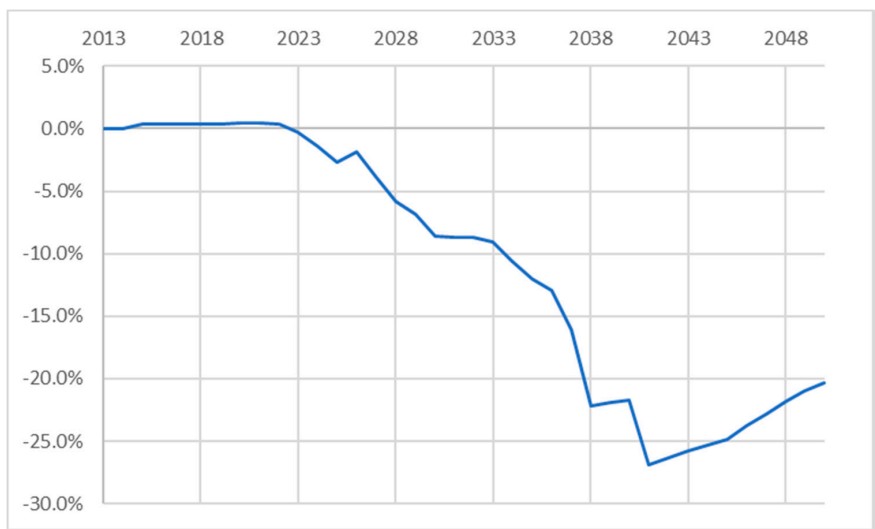

**Figure 8.** Variation in CO$_2$ Emissions Source: Own elaboration.

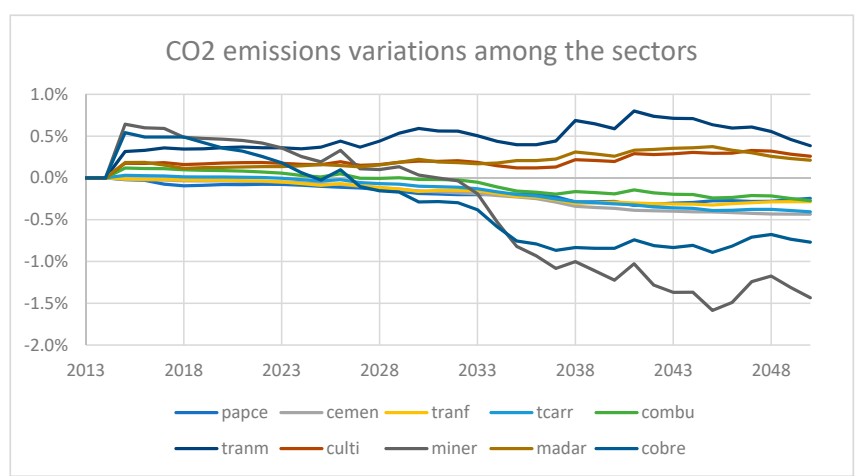

**Figure 9.** CO$_2$ emissions variations among the sectors. Source: Own elaboration.

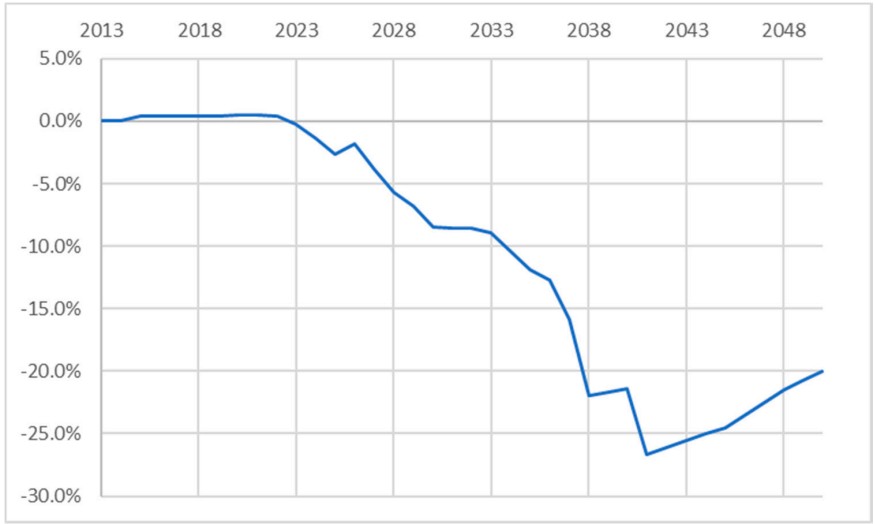

**Figure 10.** Variation in CO$_2$ emission intensity. Source: Own elaboration.

### 4.1. Impact on GDP

Figure 11 shows the difference in GDP related to a mandatory early phase-out of plants and the baseline between 2020 and 2050. A decrease can be seen as of 2017, and the greatest difference with the baseline is seen in 2024 (0.15%). After this year, the difference drops as the economy adjusts to the new restrictions. In general terms, the impact on GDP is relatively little, with a reduction of the baseline from 0.05% in 2040 to close to 0.02% by 2050. The fact that the GDP decreases is to be expected, considering the suppositions made when constructing the scenario. In particular, there is a higher cost of electricity generation for all sectors of the economy. Moreover, the economy must sub-optimally assign investment resources due to a different generation share than originally. These two factors lead to a lower GDP in this scenario. This difference will diminish as the economy adjusts to the new costs of electricity and substitutes for lower relative cost factors.

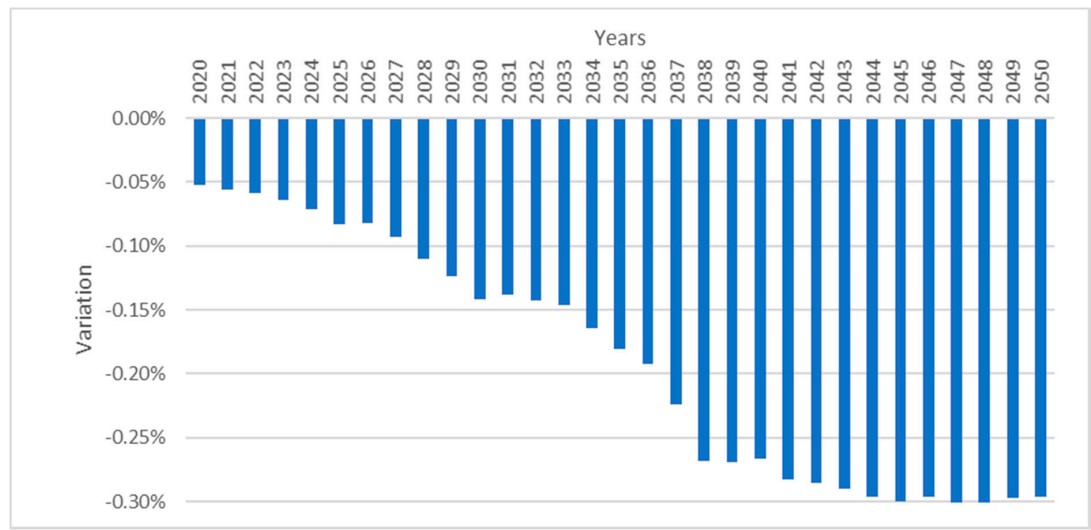

**Figure 11.** Impact on GDP. Source: Own elaboration.

### 4.2. Impact in the Labor Market

By phasing out coal plants, prices and balanced quantities in the labor market will be slightly affected. The impact on generation and the loss of jobs in the electricity generation sectors and other production sectors affected is presented below, in terms of total employment in the economy and salaries by qualification level and gender. The coal plant phase-out leads some sectors to reduce jobs and others to increase jobs. Naturally, the coal plant phase-out brings about a nearly 90% decrease—according to the Figure 12 below—in direct employment in the coal generation sector as of 2040. Due to its significance, there is also a reduction in direct jobs in the coal mining sector, whose production drops thanks to the reduction in coal-fired generation. A decrease in jobs in the coal mining sector is not as sharp as in the coal generation sector, because coal mineral is not used exclusively in coal-fired generation. Even still, by 2040, a 50% drop in jobs can be seen.

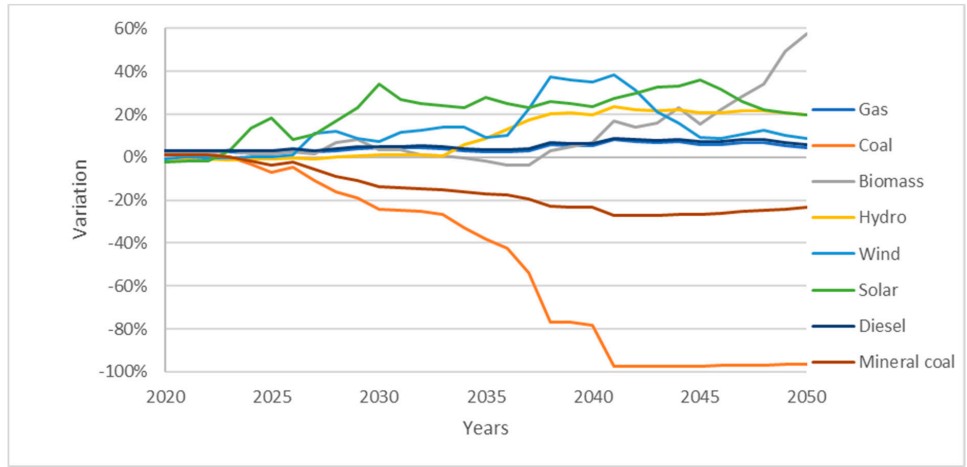

**Figure 12.** Variation in employment in sectors related to generation. Source: Own elaboration.

On the other hand, to offset the reduction in coal-fired production of electricity, other generation sectors must increase production and, therefore, their direct employment. In fact, the wind, hydro, and solar power generation sectors see an increase in share as of 2022, approximately, which brings about an increase in direct jobs in these sectors. As these sectors are based on renewable energies, they are called "green jobs."

Of course, there is also an impact on employment in other sectors not related to electricity generation, which is affected by the increase in electricity costs, the reassignment of investment resources, and a domino effect in sectors whose production varies. Figure 13 shows that there are sectors with higher levels of employment and others with lower levels. The range of variation in this case is much lower than in the sectors directly affected, with a maximum increase of 1% and maximum decrease of 3%. One sector where employment goes up is in construction, as the new generation options—and reduction in salaries as discussed below—give an impulse to this sector. On the other hand, coal generation was an important source of electricity for the metallurgy sector, therefore its disappearance implies an increase in generation costs and the operating costs for this sector. This implies a reduction in production and also in job demands. This can be seen in the following graph.

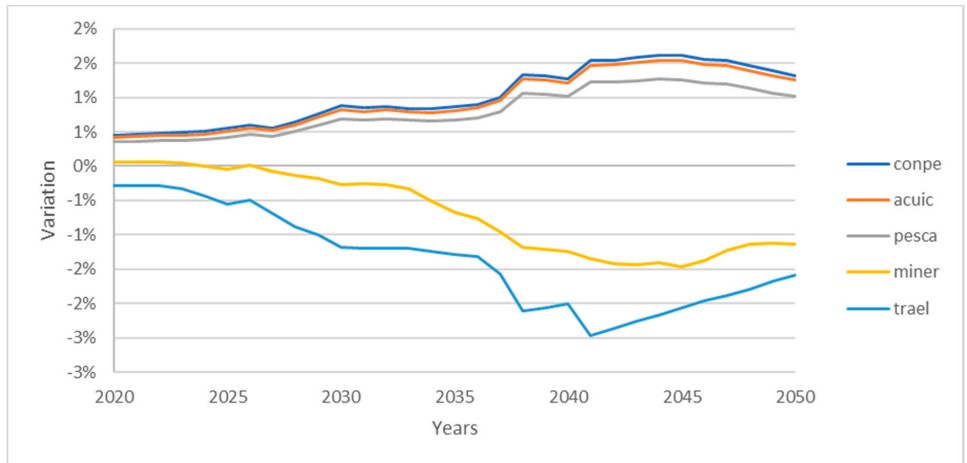

**Figure 13.** Most relevant variation in employment not related to electricity generation. Source: Own elaboration.

Total household income by quintile is also affected by the plant phase-out (see Figure 14). This income considers salary, capital, and transfers. The results shown in the figure below present

negative variations for all quintiles. The variation ranges reach their maximum around 2040, at –0.3% (for households in the second and third quintile) and stabilize between –0.1% and –0.15%. The reduction in salary income, as well as in the GDP, means that households across all five quintiles see a lower total income.

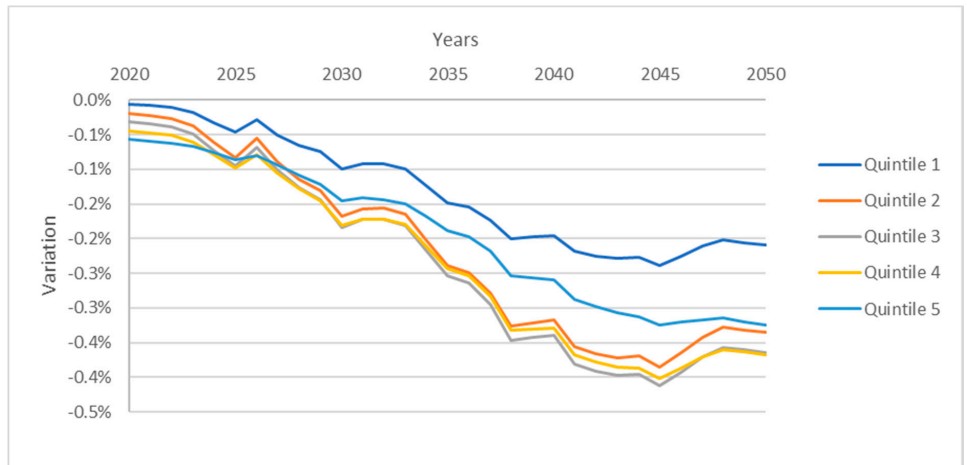

**Figure 14.** Variation in income by quintile under plant phase-out scenario. Source: Own elaboration.

### 4.3. Impact on the Copper Sector

Finally, at the sectorial level, it should be noted that copper production will be affected by a slight increase in the price of such an important input as electricity, dropping with respect to the baseline production (see Figure 15). This reduction reaches its maximum in 2024, at –0.59%, but then stabilizes at a drop of –0.3%. The coal plant phase-out scenario implies an increase in electricity prices paid by all consumers, given the new restriction imposed. Like all production sectors, copper mining is affected by the increase in electricity prices, which leads to a decrease in production. The reduction in production is summarized in the table below. Copper mining demands a large amount of electricity for production, given the machinery used in the industry. In the social accounting matrix, 54% of the sector's electricity demand comes from coal-fired generation. This may lead one to think that the decrease in production would be greater given the total phase-out of coal-fired plants. However, electricity as an input does not vary much despite the different sources from which it may come. This is reflected in the model, by using enough elasticity to represent the Chilean reality.

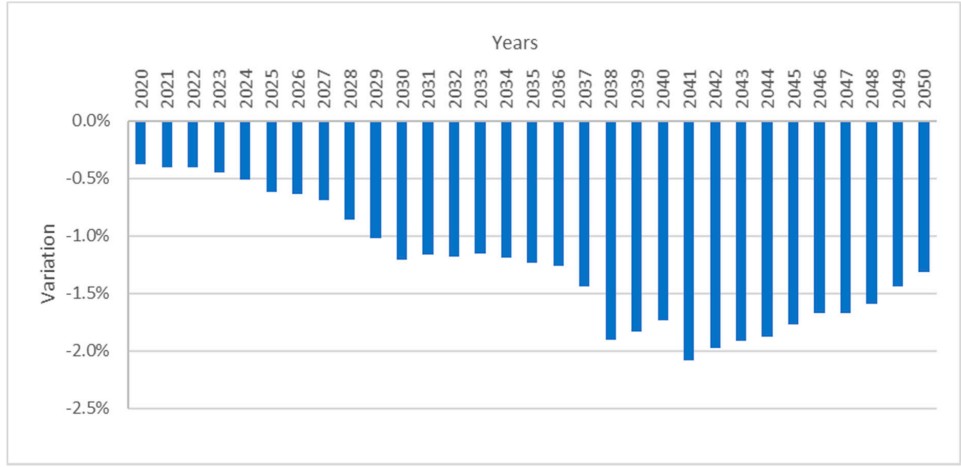

**Figure 15.** Projection of percentage variation in copper production in plant phase-out scenario with respect to the baseline. Source: own elaboration.

## 5. Discussion: Challenges to the Ambitious Decarbonization Goals

Climate change has become the greatest challenge pushing many governments to adopt several strategies to reduce GHG emissions on the short- and long-term horizons. In this context, the Chilean Government has recently established ambitious plans to eliminate some of its coal-fired power generation plants by 2024 and the rest by 2040. This pathway towards zero-carbon electricity relies heavily on development of its local rich renewable energy sources. Over the last few years, renewable energy technologies worldwide have become fundamental sources to significantly reduce energy-related carbon dioxide ($CO_2$) emissions in the power sector as they represent two-thirds of all greenhouse gases (GHG). According to a recent study by the International Renewable Energy Agency (IRENA) the expansion of renewable energy and improvement of energy efficiency can account for around 90% of energy $CO_2$ reductions [29]. To achieve this scale of decarbonization, global power generation and total primary energy supply should be represented around 80% and 65% by renewable energy sources. Although renewable energy today accounts for only 30% of the global energy generation mix, over the last few years, most of the new installed capacities in the generation mix were captured by renewable energies. In 2018, more than 60% of new power capacities have been made using these technologies. This is expected to grow even in the near future as the cost of these technologies decrease. Therefore, the levelized cost of electricity from solar photovoltaics and wind power has experienced a sharp decrease of an astounding 73% and 23% respectively between 2010 and 2017 [30]. Some renewable technologies currently in commercial use, particularly solar and wind power, are cost competitive compared to traditional fossil fuels in most regions of the world.

With one of the richest renewable energy sources, a sharp drop in technology costs, attractive market conditions and relevant policy changes, Chile has become one of the most attractive markets in the Latin American region. Chile's recent renewable energy boom has made significant improvements towards reducing emissions, and it is evident that their share in the grid will increase significantly in the near future. Given that coal power plants and NCREs currently constitute around 30% and 21% of installed capacity in the Chilean power market, respectively, the total replacement of coal plants by renewable energies by the year 2040 means that more than 60% of energy will come from variable energy sources such as solar and wind power technologies. But despite these enormous gains, it is important to determine whether or not a massive expansion in renewables in the current grid can deliver affordable, reliable, and clean energy, at least in the short- and mid-term horizons. In this decarbonization scenario for the energy sector, to what extent are the government's latest ambitious targets coherent with the current technological limitations?

Despite remarkable technological improvements related to renewable generation, storage, and transmission, there are still serious challenges that need to be highlighted and addressed through additional policy interventions so as to sustain this rapid energy transition. A first challenge is that the urgent need to replace coal plants with renewables will require significant overbuilding of renewable generation to meet its growing demand. According to long-term energy planning scenarios built by the Chilean Ministry of Energy (2018), Chile plans to develop 40% of electric vehicles and 100% electric buses by 2050. This means that the future electrification of other sectors will further increase demand for electricity. One of the important factors in the large overbuilding capacity is associated with the "capacity factor" of renewable technologies, particularly solar and wind. The average capacity factors of solar (24%) and wind (34%) technologies are much lower compared to coal-fired plants, which is 60% on average. This means that the power sector will require a larger capacity from wind or solar technologies to generate the same amount of electricity produced by coal-fired plants. Moreover, due to the variability of renewable technologies, they have a low rate of use, and most of the time, the power generated by these technologies cannot be used to meet demand. This is particularly true in the case of wind and solar power. For instance, these technologies are not available all the time and when they are, most of the time demand is low. Therefore, low capacity factors and utilization rates require cheaper solar and wind technologies to make this overbuilding capacity economically advantageous.

Another important challenge for a successful transition to a larger share of renewable energy sources in Chile is growing demand for smart grid technologies, costly grid upgrades, and a significant overbuilding of transmission infrastructure. An expansion of grid interconnection and transmission capacity is an important option for coping with the variability of wind and solar power generation, reducing the requirements for regulation reserves, improving congestion management and decreasing the need for backup generation capacity. According to the experience of countries with the highest level of renewable penetration, upgrading grid infrastructure and increasing interconnection and electricity trade have been fundamental factors for large penetration. The recent interconnection of the northern and the central power systems through a 600-km-long truck transmission was a significant step forward in Chile. This is expected to provide more flexibility in power transmission from the northern regions, making a large amount of solar power available to the central region, where there is a higher energy demand. However, over the last year, this transmission system segment has been experiencing significant congestion, leading to frequent price decoupling and curtailments [31]. As a result, the price decoupling in the case of northern Chile, which is home to rich solar resources, drove marginal costs to zero given the large penetration of renewables. Absence of strong interconnections and electricity trade with neighboring countries is an important limitation for the current and future penetration of renewables in the Chilean power system. In general, commercial exchange of electricity among South American countries lags behind other regions due to both regulatory differences and markets that function very differently. Chile only has interconnection with Argentina, and this is not currently in operation.

The deep decarbonization of the Chilean energy system can be realistic when economically and technically reliable storage technologies are available and integrated into the power system. However, these are still currently either minimal or at a very early stage of development and unlikely to sufficiently meet future flexibility requirements resulting from massive expansions of renewable technologies under decarbonization efforts. In a future low-carbon energy system, storage technologies have fundamental potential to provide a wide range of attractive benefits at all levels of the electricity system. At the generation level, they can provide arbitrage, balancing, and reserve power, etc.; at the transmission level, they can improve frequency control and investment deferral; at the distribution level, they can ensure voltage control, capacity support, etc.; and at the consumer level, they can provide peak shaving, time of use cost management reducing the volatility of market prices [32–34]. Currently, almost 99% of current established storage capacity is represented almost exclusively by pumped hydro-storage, mainly in mountainous areas. Chilean Law 20,936, first adopted in 2016, establishes the general aspects related to energy storage, defining storage systems and integrating them into power systems. Currently, regulations on storage only refer to the operation of pumping stations without hydrological variability. Regarding other battery storage technologies, although minor installations in these technologies present promising impacts on grid stability, increasing flexibility and reducing curtailments, their massive application remains limited. Over the last few years, the production of green hydrogen using renewable energy in Chile has become a high priority area for the further development of the energy sector. For this purpose, the Chilean Ministry of Energy recently recognized its ongoing work to draw up a national hydrogen strategy. Currently, fossil fuels represent almost 96% of the world's hydrogen, while the remaining 4% comes from water. However, this situation is changing quickly due to remarkable decreases in the cost of renewables, particularly solar power. Over the last few years, solar technologies in Chile have become progressively competitive due to low operating costs resulting from the high capacity factor in the northern zones with some of the best solar resources in the world. For instance, the Atacama Desert receives 3500 kWh/m$^2$ of DNI radiation and 3000 h of sunshine per year, which is 65% higher than average radiation in Europe. Therefore, this factor gives Chile a highly advantageous position to produce hydrogen at competitive prices.

## 6. Conclusions and Future Research

Chile's rich solar potential allows for the proposal of an ambitious decarbonization of its electricity generation sector that goes beyond what will happen naturally due to market conditions; however, this will impose trade-offs that must be adequately assessed. The multiple impacts of electric sector decarbonization goals relating to sustainable development requires an appropriate modelling approach. Computable general equilibrium models allow quantifying economic, social, and environmental impacts. The model must respond to the needs of policymakers, in particular replicating expected energy development pathways and allowing assessment of the policies of interest.

The dynamic recursive ECOGEM-Chile model has been adapted for this. First, the electricity generation sector has been disaggregated to include renewable energies, especially solar and wind, that are expected to be the key future generating sectors. The expected evolution of the generating sector has been soft-linked with the ECOGEM model replicating the expected future energy matrix up to 2050. The model has been modified separating the energy nesting into electricity and non-electricity sectors that have a low substitution among them. Using the model together with the best projections for key exogenous parameters such as GDP, population, and labor supply growth, a baseline scenario for the economy between 2020 and 2050 has been developed. It includes the future development of 60 economic sectors, including seven electricity generating sectors, 12 labor categories, household income per decile, and $CO_2$ emissions, as well as relevant macroeconomic variables.

Even though most of the future growth of electricity generation will come from solar and wind power, in this baseline scenario, it is still expected that 15% will be generated using coal in 2050. An alternative decarbonization scenario that considers a complete coal phase out in 2040 has been assessed considering economic, social, and environmental indicators. The results show that in this scenario, GDP can be expected to fall 0.3% in 2050 relative to the BAU scenario due to a small increase in electricity costs. This rise in electricity prices will result in a small fall in copper production of 0.3% due to the importance of electricity costs in total copper costs. On the social front, income by quintile will fall as GDP falls, by 0.2% for the poorest and between 0.3% and 0.4% for the other. In the labor market, employment in the coal electricity generation sector will fall by 100% in 2050, but green jobs in the solar and wind sectors will grow 20% and 10%, respectively. Total employment in the generating sector is not affected. Finally, $CO_2$ emissions will fall over 20% in 2050.

These results suggest that a phase-out of coal generation will have significant impacts on emissions and result in a structural shift towards green jobs, over 20%. The trade-offs, however, are that the increase in electricity costs will have negative impacts on GDP and household income that will fall around 0.3%.

**Author Contributions:** S.N., R.O., and H.O. designed and performed the research and wrote the paper with results checking. All authors read and approved the final manuscript.

**Funding:** This research was funded by ANID/FONDAP/15110019 (SERC-CHILE) and ANID/FONDECYT/ 11170424, ANID/FONDEF/ID16I10128, ANID/FONDAP/15110009.

**Conflicts of Interest:** The authors declare no conflict of interest.

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
