# Peer review of "Decarbonization Tradeoffs: A Dynamic General Equilibrium Modeling Analysis for the Chilean Power Sector"

_sustainability, doi:10.3390/su12198248_

Round 1

Reviewer 1 Report

The paper deals with the study on „Decarbonization Tradeoffs: A dynamic general equilibrium modeling analysis for the Chilean power sector”. The presented topic is of high professional and practical interest what brings a significant added value to the potential target group of readers. The overall writing style reflects a logical clear concept. Hopefully, my remarks, observations, and possible suggestions might bring the authors benefits for the enhancement of the paper to be published properly. Accordingly, I am stating my comments below.

Originality

From my point of view, the researched study can be considered as an innovative scientific contribution with clear identification of its overall structure.

Title

The title reflects the objective and content of the paper, the lenght is adequate, so no remarks related to this aspect can be pointed out.

Abstract

This part demonstrates the structured summary, background, results, conclusion, and impact of key findings.

Introduction

This part of the paper is properly designed in a correct explanatory way. This part does highlight the aims of this investigation. The most relevant part is the introduction section that gives a perfect context for the justification of the research. This section is very well based and combined with the Hypothesis Development part. The introduction is well explained and covered almost everything such as objectives, contribution, novelty, significance, and so on.

Methodology – The ECOGEM-Chile Model

I am very happy to get the opportunity to review this article. This is definitely a good piece of article. The ECOGEM-Chile macroeconomic model is well explained in the methodology section.

Results

The findings are highly focused on the objectives and significance of this study. I have one suggestion; would you please show some past data with figure 4 – 6?

I think 93 references are almost to justify this topic. But it is critical to highlight what is the gap and what is new in this study. I would encourage the authors to provide more recent references (from the last 5 years) and please delete the old references (reports). You need to develop this part. The manuscript could be even more sufficiently supported by evidence or proper references to work done elsewhere (Sustainability or other Scopus and Web of Science listed journal).  I think references are not enough to justify this topic. For this purpose authors could download the following published article:

https://www.mdpi.com/1996-1073/13/2/407

https://www.mdpi.com/1996-1073/13/10/2607

Policy prototype and assessment

Please correct in text citations “Error! Reference source not found. presents”.

Discussion

The discussion section was well written.

Conclusions

In this section the main ideas of the manuscript are presented, the obtained results and their novelty are demonstrated. The interpretations and conclusions are justified by the results.

Overall, my decision is to accept the article with minor revision

Author Response

Authors’ response to the Reviewer 1 comments

Manuscript ID: 948488

Title: Decarbonization Tradeoffs: A dynamic general equilibrium modeling analysis for the Chilean power sector

Authors: Shahriyar Nasirov  , Raúl O´Ryan and Héctor Osorio

                                                                 October 1, 2020

We thank Reviewers and Editorfor taking the time and effort necessary to revise our paper. We have studied comments carefully and have made correction which we hope meet with approval.

In what follows, we provide detailed responses to comments of the Reviewer and Editorone by one.

Reviewer Comments:

The paper deals with the study on „Decarbonization Tradeoffs: A dynamic general equilibrium modeling analysis for the Chilean power sector”. The presented topic is of high professional and practical interest what brings a significant added value to the potential target group of readers. The overall writing style reflects a logical clear concept. Hopefully, my remarks, observations, and possible suggestions might bring the authors benefits for the enhancement of the paper to be published properly. Accordingly, I am stating my comments below.

Originality:From my point of view, the researched study can be considered as an innovative scientific contribution with clear identification of its overall structure.

Title:The title reflects the objective and content of the paper, the lenght is adequate, so no remarks related to this aspect can be pointed out.

Abstract:This part demonstrates the structured summary, background, results, conclusion, and impact of key findings.

Introduction:This part of the paper is properly designed in a correct explanatory way. This part does highlight the aims of this investigation. The most relevant part is the introduction section that gives a perfect context for the justification of the research. This section is very well based and combined with the Hypothesis Development part. The introduction is well explained and covered almost everything such as objectives, contribution, novelty, significance, and so on.

Methodology – The ECOGEM-Chile Model: I am very happy to get the opportunity to review this article. This is definitely a good piece of article. The ECOGEM-Chile macroeconomic model is well explained in the methodology section.

Response: We thank the reviewer’s appreciation about our manuscript. It is great to hear about your positive opinions.

Reviewer Comments:I have one suggestion; would you please show some past data with figure 4– 6?

Response: Many thanks for your comments. Given that our macroeconomic ECOGEM -CGE model is simulating the date which was obtained from the external- engineering model, we have only considered the simulations for 2020-2050.

Reviewer Comments:I think 93 references are almost to justify this topic. But it is critical to highlight what is the gap and what is new in this study. I would encourage the authors to provide more recent references (from the last 5 years) and please delete the old references (reports). You need to develop this part. The manuscript could be even more sufficiently supported by evidence or proper references to work done elsewhere (Sustainability or other Scopus and Web of Science listed journal).  I think references are not enough to justify this topic. For this purpose, authors could download the following published article:

Response: Many Thanks for your suggestion. We have highlighted the contribution of this study in the new version. Besides, we have incorporated new references to justify the topic. Thanks for suggesting a relevant reference which we have used it in the study.

Reviewer Comments:

 Discussion: The discussion section was well written.

Conclusions: In this section the main ideas of the manuscript are presented, the obtained results and their novelty are demonstrated. The interpretations and conclusions are justified by the results.

Response: Many thanks.

Reviewer 2 Report

Sustainability Manuscript # 948488

Decarbonization Tradeoffs: A Dynamic General Equilibrium Modeling Analysis for the Chilean Power Sector

This manuscript reports on a study to determine the impacts of various decarbonization paths in the Power Sector of Chile. Given the current consumption levels of coal as an (imported) energy source in Chile, it is imperative that alternative sources be investigated rigorously. This is a very worthy study.

I am impressed with this manuscript. It is very well-written and although technical, I found it very interesting. The literature review is thorough and well organized. The signposts at the end of the Introduction are useful. The authors do an excellent job describing the role of coal plants in the context of Chile.

The methodology, while complex, is well explained to the reader. The illustrations render the models understandable. The simulations seem appropriate to answer the research question, as they consider a wide range of salient variables. The manuscripts’ conclusions follow directly from the results.

I only have one comment: the labels in Figures 8 and 11 appear to be in Spanish while the rest of the text is in English.

Author Response

Authors’ response to the Reviewer 2 comments

Manuscript ID: 948488

Title: Decarbonization Tradeoffs: A dynamic general equilibrium modeling analysis for the Chilean power sector

Authors: Shahriyar Nasirov  , Raúl O´Ryan and Héctor Osorio

                                                                 October 1, 2020

We thank Reviewers and Editorfor taking the time and effort necessary to revise our paper. We have studied comments carefully and have made correction which we hope meet with approval.

In what follows, we provide detailed responses to comments of the Reviewer and Editorone by one.

Reviewer Comments:

Decarbonization Tradeoffs: A Dynamic General Equilibrium Modeling Analysis for the Chilean Power Sector

This manuscript reports on a study to determine the impacts of various decarbonization paths in the Power Sector of Chile. Given the current consumption levels of coal as an (imported) energy source in Chile, it is imperative that alternative sources be investigated rigorously. This is a very worthy study.

I am impressed with this manuscript. It is very well-written and although technical, I found it very interesting. The literature review is thorough and well organized. The signposts at the end of the Introduction are useful. The authors do an excellent job describing the role of coal plants in the context of Chile.

The methodology, while complex, is well explained to the reader. The illustrations render the models understandable. The simulations seem appropriate to answer the research question, as they consider a wide range of salient variables. The manuscripts’ conclusions follow directly from the results.

I only have one comment: the labels in Figures 8 and 11 appear to be in Spanish while the rest of the text is in English.

Response: We thank the reviewer’s appreciation about our manuscript. It is great to hear about your positive opinions. Many thanks for your comments. We have changed them in the new version.
